# OpenReview forum: "TruthRL: Incentivizing Truthful LLMs via Reinforcement Learning"
_ICLR.cc/2026/Conference — Submitted to ICLR 2026_

### Official Review · Reviewer_a5cR · 2025-10-30

**Soundness:** 3
**Presentation:** 3
**Contribution:** 2
**Rating:** 4
**Confidence:** 4

**Summary:**

the paper proposes a novel RL framework, TruthRL, designed to directly optimize the truthfulness of LLMs.  The authors identify a critical limitation in existing methods, which either amplify hallucinations (factually incorrect outputs) when optimizing for accuracy or become overly conservative when promoting abstention. TruthRL employs a ternary reward structure that incentivizes correct responses, penalizes hallucinations, and treats abstentions neutrally. the proposed framework is evaluated on four knowledge-intensive benchmarks to validate the efficacy of the proposed framework.

**Strengths:**

1. The paper effectively identifies a critical issue in LLM behavior—hallucination versus abstention—and introduces TruthRL to address it by teaching models to abstain only when faced with out-of-knowledge questions.

2. The ternary reward structure is both simple and effective, striking a balance between rewarding correct answers, penalizing hallucinations, and treating abstentions neutrally. This approach overcomes the limitations of binary reward systems.

3. Extensive evaluations across diverse datasets, settings, and backbone models validate the effectiveness of TruthRL.

4. The paper is well-written and easy to follow.

**Weaknesses:**

1. The contribution of the paper appears incremental. The core idea is to improve factuality by leveraging the model's internal knowledge boundary, enabling it to correctly answer known questions while abstaining from unknown ones. However, prior works, such as [1] and [2], have already utilized the model's self-knowledge as a reward signal to guide it toward correctly answering known questions. The novelty here seems to lie in additionally teaching the model to abstain from answering unknown questions. It would be helpful if the authors clarified how this abstention mechanism significantly advances beyond existing approaches.

2. The training rewards are derived based on the classification of known and unknown questions, which are identified through preliminary knowledge probing on a pre-collected, task-specific annotated training dataset. This reliance on annotated datasets tailored to specific tasks could limit the approach's generalizability and applicability in real-world, dynamic environments where such datasets may not be readily available.

References:

[1] Fine-tuning Language Models for Factuality
[2] Self-Alignment for Factuality: Mitigating Hallucinations in LLMs via Self-Evaluation

**Questions:**

1. Could you provide an error analysis? Additionally, have you observed any reward hacking behavior related to the designed rewards?

2. Could you demonstrate the generalization capability of the proposed approach?

---

> ### Author Response · Authors · 2025-11-22
> **Response to Reviewer a5cR  [Part 1/3]**
>
> We sincerely thank the reviewer for the thoughtful comments.
>
> **[Q1]: The contribution of the paper appears incremental. The core idea is to improve factuality by leveraging the model's internal knowledge boundary, enabling it to correctly answer known questions while abstaining from unknown ones. However, prior works, such as [1] and [2], have already utilized the model's self-knowledge as a reward signal to guide it toward correctly answering known questions. The novelty here seems to lie in additionally teaching the model to abstain from answering unknown questions. It would be helpful if the authors clarified how this abstention mechanism significantly advances beyond existing approaches.**
>
> **[A1]**: We appreciate the reviewer’s suggestion on clarifying the differences between TruthRL and prior works, and we have added citations to [1,2] in the revised manuscript. Below we provide a detailed comparison:
> - **Unlike previous works [1,2], TruthRL does *NOT* use self-knowledge as a reward signal**. As the reviewer mentioned, both [1] and [2] rely on probing the model’s internal knowledge boundary to construct training data. In contrast, TruthRL uses only ground-truth answers as supervision and requires no prior signals derived from the model’s own knowledge. Note that, as stated in **Section 3.2**,  we do introduce a *knowledge-enhanced* variant that leverages self-knowledge to guide training, but this variant consistently underperforms our default method (**see Table 3**), demonstrating TruthRL’s effectiveness despite its simplicity.
>
> - **Moreover, the way TruthRL models abstention behavior fundamentally differs from [1] and [2]**. In DPO-based methods such as [1] and [2], abstention is encoded through **static** preference pairs that are fixed before training and cannot adapt as the policy evolves. **In contrast, TruthRL dynamically rewards or penalizes abstention through group relative advantage estimation**. The key mechanism is that, in GRPO, a response’s advantage is computed relative to the mean reward within the sampled group. With our ternary reward design, abstention is encouraged when the model fails to produce correct responses, but discouraged when correct responses appear within the same group. **Such dynamic credit assignment cannot be reproduced by fixed DPO preference pairs used in [1] and [2]**. We have updated **Section 3.2** of the paper to highlight this dynamic treatment of abstention in TruthRL.
>
> **[Q2]: The training rewards are derived based on the classification of known and unknown questions, which are identified through preliminary knowledge probing on a pre-collected, task-specific annotated training dataset. This reliance on annotated datasets tailored to specific tasks could limit the approach's generalizability and applicability in real-world, dynamic environments where such datasets may not be readily available.**
>
> **[A2]**: As clarified in **the response to Q1**, our TruthRL method does **NOT** require identifying known or unknown questions as a prior, and all training rewards in TruthRL are derived **solely by comparing the model’s prediction against the ground-truth answer**, without using any prior knowledge about whether a question is known or unknown to the model.
>
> Indeed, we fully agree with the reviewer that any reliance on pre-annotated knowledge labels could limit the method’s generalizability. This is exactly why we include a *knowledge-enhanced* variant and DPO-based baselines for comparison. As shown in **Table 3 and Table 4**, these variants consistently underperform our default TruthRL.
>
> These results demonstrate that TruthRL is both simpler and more effective than its complex counterparts. Thanks for bringing this up, we have updated **Section 3.2** to clarify this point and avoid potential confusion about our default setup.

---

> > ### Author Response · Authors · 2025-11-22
> > **Response to Reviewer a5cR  [Part 2/3]**
> >
> > **[Q3]: Could you provide an error analysis? Additionally, have you observed any reward hacking behavior related to the designed rewards?**
> >
> > **[A3]**: Thanks for the suggestion. We analyzed the predictions of different methods and identified some representative error patterns. Below, we summarize the behaviors of Prompting, R-Tuning, and TruthRL, along with a concrete example for illustration.
> > - Prompting often generates fluent but hallucinated answers when uncertain.
> > - R-Tuning reduces hallucinations by abstaining more frequently, but tends to over-abstain and fails to extract correct information when it is present.
> > - TruthRL substantially reduces hallucinations while preserving accuracy: it answers correctly when confident and abstains only when appropriate.
> >
> > > **Question**: who has had the most number one hits in country music, George Strait or Conway Twitty? \
> > > **Ground-truth Answer**: As of 2021, George Strait has the most number one hits in country music with 60. Conway Twitty has 40 number one hits. \
> > > **Prompting Prediction**:  \<think\> To answer the question, we need to determine who has had the most number one hits in country music, George Strait or Conway Twitty. \n\nFrom the given references, we can see that George Strait has had 44 number one hits, and Conway Twitty has had 40 number one hits. However, we also need to consider that Conway Twitty had 55 number one hits on various charts, including Cashbox and Radio and Records, not just Billboard. \n\nAccording to Document [40], Conway Twitty had 55 number one hits, including 55 on Cashbox and 40 on Billboard. \n\nTherefore, based on the information provided, Conway Twitty has had the most number one hits in country music, with a total of 55 number one hits. \n\nHowever, it's worth noting that George Strait has had 44 number one hits on the Billboard chart, which is a significant achievement. \n\nSo, the answer to the question is: \\boxed{Conway Twitty has had the most number one hits in country music, with a total of 55 number one hits.} \
> > > **R-Tuning Prediction**: \\boxed{I don’t know} \
> > > **TruthRL Prediction**: \<think\> To answer the question, "Who has had the most number one hits in country music, George Strait or Conway Twitty?"\n\nAccording to the references, George Strait holds the record for the most number one hits in country music with 60 number one singles, surpassing Conway Twitty's 40 number one hits. This is mentioned in multiple sources, including Document [4], Document [6], and Document [38]. \n\nTherefore, the final answer is: \\boxed{George Strait}
> >
> > We also agree that it is important to ensure that the model is not exploiting the reward mechanism, and **we do not observe any reward hacking behavior in our experiments**, as evidenced in the training dynamics (**Figure 5**).
> >
> > That said, to reassure that there is no reward hacking, we further evaluate all models using multiple independent LLM judges. The results in **Table 5** show that TruthRL consistently outperforms all baselines across different judges, indicating that the performance gain is not tied to a particular judge or simply due to reward hacking.

---

> > > ### Author Response · Authors · 2025-11-22
> > > **Response to Reviewer a5cR  [Part 3/3]**
> > >
> > > **[Q4]: Could you demonstrate the generalization capability of the proposed approach?**
> > >
> > > **[A4]**: In the main table (**Table 1**), we evaluate TruthRL on both in-domain benchmark (CRAG) and multiple out-of-domain benchmarks (NQ, HotpotQA, MuSiQue). We also validate the generalization capability of TruthRL across different model families (i.e, Llama, Qwen) and sizes (3B, 7B, 8B, 32B), as presented in **Table 8**.
> > >
> > > Per the reviewer’s suggestion, we further strengthen the evidence with additional experiments.
> > >
> > > - **TruthRL generalizes to non-QA task (GSM8K).** To test whether TruthRL generalizes beyond QA tasks, we evaluate it on the GSM8K mathematical reasoning benchmark using Llama3.1-8B-Instruct as the backbone.
> > > | **Method**     | **Truthfulness (↑)** | **Hallucination (↓)** | **Uncertain (-)** | **Accuracy (↑)** |
> > > |--------------------|:------------:|:-------------:|:---------:|:--------:|
> > > | Prompting          |     66.7     |     16.3      |   0.7     |   83.0   |
> > > | SFT                |    -67.4     |     83.7      |   0.0     |   16.3   |
> > > | RFT                |     59.8     |     19.6      |   0.9     |   79.5   |
> > > | R-Tuning           |     -4.0     |      **6.4**      |  91.3     |    2.4   |
> > > | **TruthRL** |     **71.9**     |     12.7      |   2.7     |   **84.6**   |
> > >
> > >     The results show that TruthRL maintains strong truthfulness and low hallucination while preserving high accuracy, showing that the proposed method generalizes beyond QA tasks.
> > >
> > > - **TruthRL generalizes well to production-level LLMs and achieves comparable results with GPT-5**. We also apply TruthRL to Llama3.3-70B-Instruct and compare it against strong proprietary models such as GPT-5 and OpenAI o3. The results show that the 70B model trained with TruthRL achieves promising truthfulness comparable to highly capable production systems.
> > > | **Method**     | **Truthfulness (↑)** | **Hallucination (↓)** | **Uncertain (-)** | **Accuracy (↑)** |
> > > |------------------------|:------------:|:-------------:|:---------:|:--------:|
> > > | GPT-5                  |     **42.3**     |     24.7      |   8.2     |   **67.0**   |
> > > | o3                     |     33.2     |     33.1      |   0.6     |   66.3   |
> > > | **TruthRL (70B)**      |     40.6     |     **17.3**      |  24.9     |   57.9   |
> > >
> > > Overall, these additional experiments across task types and model scales confirm the generalization capability of TruthRL.
> > >
> > > Please let us know if you have any further questions, and we are happy to incorporate additional suggestions you might have! If you find our response satisfactory, we would be grateful if you could consider raising your score. Thanks again for your time and efforts!
> > >
> > > **References**
> > >
> > > [1]. Tian et al. Fine-Tuning Language Models for Factuality. ICLR 2024.
> > >
> > > [2]. Zhang et al. Self-Alignment for Factuality: Mitigating Hallucinations in LLMs via Self-Evaluation. ACL 2024.

---

### Official Review · Reviewer_JATC · 2025-10-30

**Soundness:** 3
**Presentation:** 3
**Contribution:** 2
**Rating:** 4
**Confidence:** 4

**Summary:**

This paper proposes TruthRL, an RL framework designed to improve the truthfulness of LLMs. It proposes a simple and effective ternary reward scheme to encourage the model to perform correct abstention and reduce hallucination.

**Strengths:**

1. Comprehensive experimental results and thorough analysis. This work includes a substantial number of experiments and appears to have been optimized multiple times. Some conclusions are insightful, such as the comparison between binary and ternary rewards, and the ineffectiveness of offline-RL.
2. Good presentation and writing. The authors' writing is clear, and the figures and tables are well-drawn and concise.

**Weaknesses:**

1. The method's novelty is insufficient, and the chosen baselines are not strong enough. The field of LLM reliability and abstention already has a considerable number of papers. Similar ternary rewards have been proposed in previous work, such as [1][2]. Especially in [2], the authors discussed in detail how the reward value for abstention in a ternary reward scheme affects model behavior; it is recommended to reference this. Furthermore, the baselines selected in this work only include SFT-based methods. There are many more novel methods that could potentially achieve better performance, such as [1][3][4]. Some of these are already mentioned in the related work, and maybe the authors should consider including them as baselines.
2. Metric Definition: The chosen Truthfulness metric, defined as T=Acc - Hall, is a relatively basic metric that lacks consideration for abstention. A problematic scenario might arise: a model that can make mistakes (Acc=80, Unc=0, Hall=20, T=60) and a model that is perfectly safe (Acc=60, Unc=40, Hall=0, T=60) would receive the same Truthfulness score. The field of model reliability already has many more sophisticated metrics, which can be referenced in [5].

References

[1] https://arxiv.org/abs/2403.18349

[2] https://arxiv.org/abs/2403.05612

[3] https://arxiv.org/abs/2502.07184

[4] https://arxiv.org/abs/2412.11803

[5] https://arxiv.org/abs/2407.18418

**Questions:**

1. About training stability: I have also explored this simple ternary reward setting for training abstention and found that the RL training is not stable. The model's behavior tends to collapse towards either always answering or always refusing as training progresses. Using this ternary reward with the GRPO algorithm might also pose a problem. As training continues, the model will likely exhibit singular actions within the sampled group for most queries (i.e., either always answering or always refusing), even with a high temperature T=1. This leads to uniform rewards within the same query group (e.g., all 1s or all 0s), which is detrimental for GRPO's calculation of the in-group advantage and further optimization. I would like to know if you also experienced instability during your training process, and if you observed a similar phenomenon of highly consistent sampled actions. Maybe using other RL algorithms that do not rely on in-group sampling could further improve the training effectiveness.
2. About offline DPO: The performance of offline DPO appears to be very poor. Have you conducted further research into why its performance is so low? Perhaps it is an issue with the training data construction or the data-mix ratio? According to results from [1], they also employed a similar offline DPO method but used a different data construction strategy, achieving relatively good performance.

References

[1] https://arxiv.org/abs/2403.18349

---

> ### Author Response · Authors · 2025-11-22
> **Response to Reviewer JATC [Part 1/3]**
>
> We sincerely appreciate the reviewer's insightful feedback.
>
> **[Q1]: The method's novelty is insufficient, and the chosen baselines are not strong enough. Similar ternary rewards have been proposed in previous work, such as [1][2]. Especially in [2], the authors discussed in detail how the reward value for abstention in a ternary reward scheme affects model behavior; it is recommended to reference this. Furthermore, the baselines selected in this work only include SFT-based methods. There are many more novel methods that could potentially achieve better performance, such as [1][3][4]. Maybe the authors should consider including them as baselines.**
>
> **[A1]**: We appreciate the reviewer’s suggestions. We have already cited [1,2] in the original submission and have added citations for [3,4] in the revised version. Regarding comparisons to prior works that adopt ternary rewards, we would like to clarify the following distinctions:
> - RLKF [1] uses a ternary reward only for their heuristic rule-based PPO as a baseline. This baseline is not the core contribution in [1] and significantly underperforms their proposed RLKF, which does not use the ternary reward but instead requires training a separate reward model. **Thus, the usage of ternary rewards in [1] is neither central to the method nor shown to be effective on its own**. Moreover, our TruthRL is also fundamentally different from RLKF: RLKF depends on offline preference data, a trained reward model, and uses pattern matching to judge accuracy, while TruthRL employs on-policy update, ground-truth based rewards, and LLM-based judge, resulting in a simpler pipeline with more reliable rewards and better suitability for free-form generation tasks.
> - **[2] also employs a ternary reward, but the focus is on studying the model behaviors under various familiarity levels of test samples, not on improving LLM truthfulness**. Their results are mostly qualitative, lacking large-scale quantitative evaluations that compare the truthfulness of various training methods across multiple QA datasets. In contrast, TruthRL provides a systematic comparison between binary and ternary reward schemes, showing that the binary scheme yields higher accuracy, while the ternary scheme is substantially more effective at improving truthfulness.
>
> Moreover, we would like to highlight that the ternary reward scheme in TruthRL is non-trivial particularly when used with RL algorithms based on group-relative advantage estimation, and **this unique dynamic credit assignment for abstention has not been studied in prior works**. For example, consider a group consisting of two responses $y_1$ and $y_2$ where $y_1$ expresses *abstention* and $y_2$ contains *hallucination*.
>  - Under the *binary* reward scheme, both responses receive a $-1$ reward (i.e., $r_{\text{binary}}(x, y_1) = r_{\text{binary}}(x, y_2) = -1$), yielding a zero relative advantage (i.e., $\hat{A}\_{\text{binary}}(x, y_1) = \hat{A}\_{\text{binary}}(x, y_2)$)---thus the policy update conflates hallucination with abstention.
> - Under the *ternary* reward scheme, $y_1$ receives a $0$ reward while $y_2$ receives a $-1$ reward (i.e., $r_{\text{ternary}}(x, y_1) = 0,  r_{\text{ternary}}(x, y_2) = -1$), resulting in a larger advantage for abstention than for hallucination (i.e., $\hat{A}\_{\text{ternary}}(x, y_1) > \hat{A}\_{\text{ternary}}(x, y_2)$). This encourages the models to abstain rather than hallucinate when they lack the knowledge to make accurate predictions.
>
> This exemplifies how TruthRL inherently better distinguishes abstention from hallucination with the relative advantage estimation of GRPO. We’ve updated **Section 3.2** to highlight this.
>
> **Per the reviewer’s suggestion, we have also added more baselines to further validate the effectiveness of TruthRL**. Specifically, we reproduced two strong RL-based methods, RLHF and RLKF, as introduced in [1]. Please refer to **Table 1 in our updated PDF** for the complete results.

---

> > ### Author Response · Authors · 2025-11-22
> > **Response to Reviewer JATC [Part 2/3]**
> >
> > **[Q2]: Metric Definition: The chosen Truthfulness metric, defined as T=Acc - Hall, is a relatively basic metric that lacks consideration for abstention. A problematic scenario might arise: a model that can make mistakes (Acc=80, Unc=0, Hall=20, T=60) and a model that is perfectly safe (Acc=60, Unc=40, Hall=0, T=60) would receive the same Truthfulness score. The field of model reliability already has many more sophisticated metrics, which can be referenced in [5].**
> >
> > **[A2]**: While our truthfulness metric follows standard practice in prior works [6,7,8], we fully agree that there are multiple ways to measure truthfulness/reliability, and that no single metric captures every aspect of model behavior.
> >
> > As suggested, we additionally evaluate all models using two additional reliability metrics introduced in [5], including **Reliable Accuracy**, and **Effective Reliability**. These metrics evaluate the model from various perspectives and allow for a more comprehensive comparison.
> > | **Method**   | **Reliable Acc. (↑)** | **Effective Reliability (↑)** | **Average (↑)**|
> > |------------------|:-------------------:|:---------------------------:|:------------:|
> > | Prompting      | 52.9               | 5.3                          |  50.2  |
> > |SFT                  |   50.7            | 1.4                          | 50.7   |
> > |RFT                   | 48.0              | -3.7                         | 46.1   |
> > | R-Tuning          | 59.4              | 15.2                         |  52.0  |
> > | TruthRL (Binary)  | 60.4               | 20.8                          |   60.3   |
> > | TruthRL              | **74.4**         | **37.2**                  | **62.5**  |
> >
> > ​​Across all reliability metrics, our method TruthRL achieves the highest overall performance, demonstrating that our conclusions remain consistent under a broader set of reliability metrics.
> >
> >
> > **[Q3]: About training stability: I have also explored this simple ternary reward setting for training abstention and found that the RL training is not stable. The model's behavior tends to collapse towards either always answering or always refusing as training progresses. I would like to know if you also experienced instability during your training process, and if you observed a similar phenomenon of highly consistent sampled actions.**
> >
> >
> > **[A3]**: We appreciate the reviewer’s insightful comments and for sharing their prior experience with ternary-reward RL training. We agree that training stability is a critical factor, and we carefully investigated training dynamics in our experiments.
> >
> > As shown in **Figure 5** of our latest paper, the training dynamics of TruthRL remain stable for both the binary-reward and ternary-reward variants. Specifically, with the ternary reward, the model produces quite balanced and stable behavior throughout training. **We do not observe degeneracy toward all-answering or all-refusing**. Instead, the model maintains a healthy distribution of answer/abstain actions over the course of training.
> >
> >
> >
> >
> >
> > **Nonetheless, we did observe training instability with ternary rewards when using heuristic rules to provide training signals.** As shown in **Table 7** in the Appendix, when we replace the LM-based verifier with a rule-based verifier (i.e., exact string matching), training quickly collapses and the model converges to almost always abstaining.
> >
> > This instability arises because heuristic rules fail to handle semantic equivalence and frequently misjudge the correctness of predictions that align with ground-truths but are expressed in different forms. Consequently, many correct predictions receive negative rewards. **When most sampled responses fall into the same negative category, the model is heavily penalized and is pushed toward responding “I don’t know” to avoid further penalties.** This highlights that reliable reward signals are crucial for stable RL training, and is precisely why TruthRL adopts an LM-based verifier instead of a rule-based one.

---

> ### Author Response · Authors · 2025-11-22
> **Response to Reviewer JATC [Part 3/3]**
>
> **[Q4]: About offline DPO: The performance of offline DPO appears to be very poor. Have you conducted further research into why its performance is so low? Perhaps it is an issue with the training data construction or the data-mix ratio? According to results from [1], they also employed a similar offline DPO method but used a different data construction strategy, achieving relatively good performance.**
>
> **[A4]**: We would like to clarify that the method used in RLKF [1] is not offline DPO. **Instead, the proposed method in [1] follows a full RLHF-style (i.e., online RL) pipeline**: it requires (i) a reward model, (ii) a policy model, and (iii) a critic model, and it uses PPO for optimization. This setup involves complex data construction and reward modeling training, which is fundamentally different from the offline DPO baseline. **In contrast, the offline DPO method trains the policy model entirely on offline preference pairs** constructed from model responses, with no online updates or interaction with a reward model.
>
> As discussed in **Section 4.4**, we attribute the weak performance of offline DPO to the fact that **offline DPO cannot learn nuanced abstention behavior due to the static nature of preference pairs**. Specifically, in offline DPO:
> - For OOK questions, preference pairs are limited to *abstention > hallucination*.
> - For non-OOK questions, the preference pairs model *correct > abstention* and *correct > incorrect*.
>
> These preferences are fully determined *before* training begins and remain static during training, thus **the offline data cannot reflect when to abstain versus answer as the policy model evolves**. We also experimented with different data-mix ratios for different types of pairs, but did not observe significant improvements.
>
> To confirm our understanding of offline DPO, we also explored **iterative DPO**, where preference pairs are regenerated for multiple rounds using the latest policy. As shown in **Table 4**, while this semi-online paradigm alleviates the limitations of offline DPO, its performance still lags significantly behind fully online RL methods like our TruthRL. **This observation is consistent with prior findings in the literature [9]**, which also demonstrates that offline and semi-online DPO variants are generally less effective than fully online RL methods.
>
> Please let us know if you have any further questions, and we are happy to incorporate additional suggestions you might have! If you find our response satisfactory, we would be grateful if you could consider raising your score. Thanks again for your time and efforts!
>
>
> **References**
>
> [1]. Xu et al. Rejection Improves Reliability: Training LLMs to Refuse Unknown Questions Using RL from Knowledge Feedback. COLM 2024.
>
> [2]. Kang et al. Unfamiliar Finetuning Examples Control How Language Models Hallucinate. NAACL 2025.
>
> [3]. Li et al. Refine Knowledge of Large Language Models via Adaptive Contrastive Learning. ICLR 2025.
>
> [4]. Xue et al. UAlign: Leveraging Uncertainty Estimations for Factuality Alignment on Large Language Models. ACL 2025.
>
> [5]. Wen et al. Know Your Limits: A Survey of Abstention in Large Language Models. TACL 2025.
>
> [6]. Yang et al. CRAG -- Comprehensive RAG Benchmark.  NeurIPS 2024.
>
> [7]. Huang et al. ConfQA: Answer Only If You Are Confident. arXiv:2506.07309
>
> [8]. Kachuee et al. PrismRAG: Boosting RAG factuality with distractor resilience and strategized reasoning. EMNLP 2025.
>
> [9]. Lanchantin et al. Bridging Offline and Online Reinforcement Learning for LLMs. arXiv:2506.21495

---

### Official Review · Reviewer_Q5vx · 2025-10-31

**Soundness:** 3
**Presentation:** 3
**Contribution:** 2
**Rating:** 4
**Confidence:** 4

**Summary:**

The paper introduces TruthRL, an online reinforcement-learning framework (implemented with GRPO) that optimizes a truthfulness objective rather than pure accuracy. The core technical idea is a ternary reward (correct = +1, abstain = 0, hallucination = −1), with optional knowledge-enhanced and reasoning-enhanced variants. The authors evaluate TruthRL on four knowledge-intensive benchmarks (CRAG, NQ, HotpotQA, MuSiQue) in retrieval and non-retrieval settings, compare to SFT, RFT, R-Tuning and RL baselines, and report that TruthRL reduces hallucinations and increases a composite truthfulness score while preserving competitive accuracy . They also analyze behavior across confidence buckets and with different judges.

**Strengths:**

Experiments cover multiple datasets, backbones (3B→32B), retrieval/no-retrieval settings and several RL paradigms, which gives a good picture of where the approach works and where it doesn’t. The paper includes analyses of confidence buckets, hallucination-baiting questions, and judge robustness, which help understand the mechanisms behind performance gains.

**Weaknesses:**

1. Lack of novelty and related works awareness. The main conceptual contribution—training models to abstain appropriately and using richer reward structures to encourage calibrated abstention—has strong prior art, which is a fatal issue. In particular: Reinforcement frameworks that encourage abstention have already been proposed (https://arxiv.org/abs/2403.18349), which used exactly the same ternary reward to teach refusal and increase reliability. The claim that the innovation is new and the “main contribution” is the ternary reward + GRPO is overstated. The authors must clearly and explicitly position TruthRL relative to these prior approaches: what is strictly new that prior work did not do?
2. Lack of ablation on reward weights. The truthfulness metric uses fixed weights (w1=1, w2=0, w3=1). Why set Unc weight w2 = 0 in experiments if abstention is a desired property? How sensitive are results to the weighting choice? Provide ablations for the weighting and for scaling of the ternary reward.

**Questions:**

1. Can you provide a human annotation on a representative sample to confirm the LLM judge labels ? If humans disagree with the LLM judge systematically, how does that affect the results?
2. How sensitive are results to the OOK detection threshold (sampling 256 responses)? If experiments with 128/512 sampling, how the number of OOKs and downstream TruthRL performance change?

---

> ### Author Response · Authors · 2025-11-22
> **Response to Reviewer Q5vx [Part 1/3]**
>
> We sincerely appreciate the reviewer’s constructive suggestions, which have significantly improved our paper. We have added discussions to better position our method and completed all suggested experiments and analysis.
>
> **[Q1]: Lack of novelty and related works awareness. The authors must clearly and explicitly position TruthRL. What is strictly new that prior work did not do?**
>
> **[A1]**: Thanks for the suggestion on clarifying the difference between our TruthRL and RLKF [1]. We have cited this work in our original submission, and below is a detailed comparison:
>
> - RLKF requires an offline preference dataset collection based on the initial model checkpoint. However, as RLKF training progresses and the policy model deviates from the initial checkpoint, the knowledge feedback signals reflected from the offline dataset may no longer be accurate for the latest policy model. **In contrast, TruthRL always updates the policy model based on its current behaviors (e.g., encouraging or penalizing abstention based on its latest knowledge)**. We’ve updated **Section 3.2** to make this clearer.
>
> - RLKF requires training a separate reward model to generate rewards for model responses, which not only complicates the overall framework but also introduces the risk of reward hacking (e.g., the policy model learns spurious patterns captured by the reward model instead of ground-truth knowledge). **In contrast, TruthRL always generates reward signals by directly comparing the model responses and ground-truth labels, without the need for an additional reward model.**
>
> - RLKF relies on pattern matching (i.e., exact match) to judge accuracy, which is not applicable to free-form generation tasks. **In contrast, TruthRL employs a strong LLM-based verifier to provide reliable correctness judgments**. Please also refer to **our response to Q3** for an example that can hardly be properly judged by exact match, but can be easily handled by an LLM-based verifier.
>
> - Regarding the use of a ternary reward, we would like to note that the ternary reward scheme is only used for their heuristic rule-based PPO, not for the proposed method RLKF. This baseline is not a core contribution of RLKF and significantly underperforms their proposed method. **Thus, the usage of ternary rewards in RLKF is neither central to the method nor shown to be effective on its own.**
>
> **Empirically, TruthRL significantly outperforms RLKF across all datasets and backbone models**. In addition to the above conceptual differences, we also conduct an empirical comparison between RLKF and TruthRL, which further confirms the effectiveness of our method despite its simplicity. Please refer to **Table 1 in our updated PDF** for the complete results.

---

> ### Author Response · Authors · 2025-11-22
> **Response to Reviewer Q5vx [Part 2/3]**
>
> **[Q2]: Lack of ablation on reward weights. Provide ablations for the weighting and for scaling of the ternary reward.**
>
> **[A2]**: We would first like to clarify that our definition of the truthfulness score (also known as effective reliability) is widely used and consistent with prior literature [2, 3, 4, 5, 6, 7].
>
> That said, we agree that the abstention weight in the truthfulness metric may vary depending on application needs, and it is indeed valuable to investigate how different weightings in the ternary reward affect model performance.
> ### **Ablation 1: Training TruthRL with different abstention rewards**
> As suggested, we ablate the abstention reward during training by setting it to negative (-0.5), neutral (0), and positive (0.5):
> | **Method**    | **Truthfulness (↑)**  |**Hallucination (↓)**|**Uncertain (-)** | **Accuracy (↑)** |
> |----------------------|:---------------------------:|:------------------------:|:--------------------------:|:--------------------------:|
> | TruthRL (negative)  | 30.8  | 26.9            | 15.4      | **57.7**       |
> | TruthRL (neutral)    | **37.2**   | 19.4            | 24.0      | 56.6       |
> | TruthRL (positive)   | 34.7     | **15.8**            | 33.7      | 50.5       |
>
> ### **Ablation 2: Evaluating truthfulness with different abstention weights**
> We further evaluate each trained model using three different truthfulness metrics corresponding to abstention weights $w_2 \in$ {-0.5, 0, 0.5}.
> | **Method**               | **Truthfulness ($w_2 = -0.5$)**|**Truthfulness ($w_2= 0$)**|**Truthfulness ($w_2 = 0.5$)**|
> |----------------------|:---------------------------:|:------------------------:|:--------------------------:|
> | Prompting            | 1.5                       | 5.3                    | 9.2                      |
> | SFT                  | 1.4                       | 1.4                    | 1.4                      |
> | RFT                  | -6.8                      | -3.7                   | -0.7                     |
> | R-Tuning             | 5.9                       | 15.2                   | 24.5                     |
> | TruthRL (negative)   | 23.1                      | 30.8                   | 38.5                     |
> | TruthRL (neutral)    | **25.2**                  | **37.2**               | 49.2                 |
> | TruthRL (positive)   |        17.9               | 34.7                   |          **51.6**            |
>
> The above results confirm that TruthRL consistently outperforms all baselines by a large margin under various scenarios, demonstrating its robustness to the choice of reward weights.
>
> **[Q3]: Can you provide a human annotation on a representative sample to confirm the LLM judge labels ? If humans disagree with the LLM judge systematically, how does that affect the results?**
>
> **[A3]**:  We agree that it is important to ensure reliable judge labels. Indeed, our work follows the same LLM-judge setup used in CRAG [2], which is **highly reliable with over 99% agreement** between LLM-judge labels and human annotations (see **Table 10 of CRAG**). Please also refer to **Table 11 of our paper** for the LLM-as-a-judge prompt.
>
> Unlike judging open-ended generation tasks where ground truth is unavailable, the judgment task in our work simply compares the predicted answer against a ground-truth reference. This is relatively straightforward for modern LLMs, making systematic disagreement unlikely. We also manually inspected a subset of LLM-judge outputs in our experiments and found them to be very reliable. Below is a representative example illustrating the accuracy of the LLM-judge, and we also added this example to the appendix of our updated paper (**Table 13**).
>
> > **Question**: How many times has Lizzo won the Grammy Award for Album of the Year? \
> > **Ground Truth Answer**: Lizzo has never won the Grammy Award for Album of the Year, though she has been nominated twice. \
> > **Predicted Answer**: \<think\> To answer the question "how many times has Lizzo won the Grammy Award for Album of the Year?", we need to analyze the information provided in the references. Document [4] mentions that Lizzo has won 4 Grammy Awards, but it does not specifically mention the category of Album of the Year. However, Document [5] mentions that Lizzo's album "Special" is nominated for Album of the Year, but it does not mention that she has won the award. After reviewing all the documents, I found that none of them explicitly state that Lizzo has won the Grammy Award for Album of the Year. Therefore, the answer is 0, as there is no evidence to suggest that Lizzo has won the Grammy Award for Album of the Year. \</think\>\n\n\\boxed{0} \
> > **LLM judge results**: {"score": 1, "explanation": "The prediction correctly states the number of times Lizzo has won the Grammy Award for Album of the Year, which is zero."}

---

> > ### Author Response · Authors · 2025-11-22
> > **Response to Reviewer Q5vx [Part 3/3]**
> >
> > **[Q4]: How sensitive are results to the OOK detection threshold (sampling 256 responses)? If experiments with 128/512 sampling, how the number of OOKs and downstream TruthRL performance change?**
> >
> > **[A4]**: We would like to clarify that **TruthRL does not rely on OOK detection—only the *knowledge-enhanced variant* uses an OOK set to provide training signals**, as stated in **Section 3.2**. Moreover, the results in **Table 3** also show that this variant consistently underperforms our default TruthRL trained with a simple ternary reward, demonstrating the effectiveness of TruthRL despite its simplicity.
> >
> > Following the reviewer’s suggestion, we vary the OOK detection process by sampling 128/256/512 responses, and report (i) the number of identified OOK questions and (ii) downstream performance of the knowledge-enhanced variant on CRAG.
> > | **Method**   | **# of OOKs**    | **Truthfulness (↑)** | **Hallucination (↓)** | **Uncertain (-)** | **Accuracy (↑)** |
> > |----------------------------------------|:--------------:|:--------------:|:------------------:|:-----------:|:------------:|
> > | Knowledge-Enhanced TruthRL (N = 128)   | 111 | 30.6   | 21.5  | 26.4      | 52.1   |
> > | Knowledge-Enhanced TruthRL (N = 256)   | 78   | 32.7   | 21.9   | 23.5      | 54.6       |
> > | Knowledge-Enhanced TruthRL (N = 512)   | 75  | 32.9    | 22.2        | 22.7      | 55.1       |
> > | **TruthRL (without Knowledge-Enhancement)**| N/A     | **37.2** | **19.4**  | 24.0 | **56.6**   |
> >
> > These results show that increasing the sampling size leads to fewer questions being marked as OOK. **More importantly, the knowledge-enhanced variant consistently underperforms the default TruthRL**. This is likely because OOK detection is performed using the initial model checkpoint before training, and the knowledge feedback from this pre-identified OOK set may no longer be accurate for the latest policy model as the training progresses. The above result aligns with our findings in **Table 3** and further **confirms that the simple ternary reward is already effective without requiring any additional complexities.**
> >
> >
> > Please let us know if you have any further questions, and we are happy to incorporate additional suggestions you might have! If you find our response satisfactory, we would be grateful if you could consider raising your score. Thanks again for your time and efforts!
> >
> > **References**
> >
> >
> > [1]. Xu et al. Rejection Improves Reliability: Training LLMs to Refuse Unknown Questions Using RL from Knowledge Feedback. COLM 2024.
> >
> > [2]. Yang et al. CRAG -- Comprehensive RAG Benchmark.  NeurIPS 2024.
> >
> > [3]. Huang et al. ConfQA: Answer Only If You Are Confident. arXiv:2506.07309
> >
> > [4]. Kachuee et al. PrismRAG: Boosting RAG factuality with distractor resilience and strategized reasoning. EMNLP 2025.
> >
> > [5] Feng et al. Don’t hallucinate, abstain: Identifying LLM knowledge gaps via Multi-LLM collaboration. ACL 2024.
> >
> > [6]. Whitehead et al. Reliable visual question answering: Abstain rather than answer incorrectly. ECCV 2022.
> >
> > [7]. Wen et al. Know Your Limits: A Survey of Abstention in Large Language Models. TACL 2025.

---

### Official Review · Reviewer_uW9S · 2025-11-02

**Soundness:** 2
**Presentation:** 3
**Contribution:** 3
**Rating:** 6
**Confidence:** 3

**Summary:**

The paper proposes TruthRL, an RL framework to directly optimize truthfulness of LLMs rather than accuracy. It uses a ternary reward, encouraging the model to both answer correctly and abstain when uncertain. Implemented via GRPO, extensive experiments demonstrate TruthRL significantly reduces hallucinations and improves truthfulness.

**Strengths:**

* The motivation is clear: addressing the accuracy–truthfulness tradeoff directly via reward shaping
* The method is easy to follow and the presentation is good.

**Weaknesses:**

* The evaluation is restricted to QA benchmarks. While the proposed ternary reward effectively balances accuracy and abstention in such settings, it remains unclear whether TruthRL generalizes to other task types, such as mathematical reasoning, or instruction-following, where uncertainty and partial correctness behave differently.
* The method can be viewed as a principled extension of reward shaping; the algorithmic innovation is limited.

**Questions:**

None

---

> ### Author Response · Authors · 2025-11-22
> **Response to Reviewer uW9S**
>
> We thank the reviewer for the positive feedback and the thoughtful comments.
>
> **[Q1]: The evaluation is restricted to QA benchmarks. It remains unclear whether TruthRL generalizes to other task types.**
>
> **[A1]**: We would like to note that QA benchmarks are widely used in this line of research for evaluating truthfulness [1,2,3], where hallucination is a critical issue and abstention plays an important role in ensuring truthfulness.
> That said, we fully agree that evaluating other task types would further strengthen the evidence for TruthRL. In light of this, we provide additional evaluation results on a mathematical reasoning benchmark (GSM8K).
> | **Method**     | **Truthfulness (↑)** | **Hallucination (↓)** | **Uncertain (-)** | **Accuracy (↑)** |
> |--------------------|:------------:|:-------------:|:---------:|:--------:|
> | Prompting          |     66.7     |     16.3      |    0.7    |   83.0   |
> | SFT                |      -67.4       |      83.7       |   0     |   16.3     |
> | RFT                |      59.8       |     19.6        |   0.9     |  79.5     |
> | R-Tuning           |     -4.0     |      6.4      |   91.3    |   2.4    |
> | **TruthRL**            |     **71.9**     |     **12.7**      |    **2.7**    | **84.6**    |
>
> The results confirm that TruthRL continues to improve truthfulness and reduce hallucinations while maintaining high accuracy for math tasks, suggesting that the method generalizes beyond QA-style benchmarks.
>
> **[Q2]: The method can be viewed as a principled extension of reward shaping; the algorithmic innovation is limited.**
>
> **[A2]**: We appreciate the reviewer’s thoughtful comments. Indeed, TruthRL highlights the effectiveness of a simple yet principled ternary reward scheme for incentivizing truthful LLMs.  Nonetheless, we would like to clarify that the innovation in TruthRL goes beyond reward design and fundamentally differs from typical reward shaping.
>
> **Algorithmically, the ternary reward scheme in TruthRL is non-trivial under RL algorithms with group relative advantage estimation (e.g., GRPO) and yields behavior that cannot be obtained through reward shaping on binary rewards.** As examplified in our **updated Section 3.2**, consider a group consisting of two responses $y_1$ and $y_2$ where $y_1$ expresses *abstention* and $y_2$ contains *hallucination*.
> - Under the *binary* reward scheme, both responses receive a $-1$ reward (i.e., $r_{\text{binary}}(x, y_1) = r_{\text{binary}}(x, y_2) = -1$), yielding a zero relative advantage (i.e., $\hat{A}\_{\text{binary}}(x, y_1) = \hat{A}\_{\text{binary}}(x, y_2)$)---thus the policy update conflates hallucination with abstention.
> - Under the *ternary* reward scheme, $y_1$ receives a $0$ reward while $y_2$ receives a $-1$ reward (i.e., $r_{\text{ternary}}(x, y_1) = 0,  r_{\text{ternary}}(x, y_2) = -1$), resulting in a larger advantage for abstention than for hallucination (i.e., $\hat{A}\_{\text{ternary}}(x, y_1) > \hat{A}\_{\text{ternary}}(x, y_2)$). This encourages the models to abstain rather than hallucinate when they lack the knowledge to make accurate predictions.
>
> This exemplifies how TruthRL differs from typical reward shaping not only in learning objective, but also inherently better distinguishes abstention from hallucination with the relative advantage estimation of GRPO.
>
>
> **As a result, TruthRL induces fundamental differences in learning dynamics**. Empirically, we observe that models trained with binary rewards quickly suppress abstention behaviors to nearly zero, whereas models trained with the ternary reward scheme maintain a stable and meaningful level of abstention, as shown in **Figure 3(a) of the updated paper**. To further understand this effect, we categorize GRPO sampling groups into two types:
> - **Pro-Abstention groups where the model is encouraged to abstain** (i.e., groups containing only incorrect and abstention responses; no correct response): As shown in **Figure 3(b)**,  the abstention ratio consistently increases across training under the ternary reward.
> - **Anti-Abstention groups where abstention is penalized** (i.e., groups that include abstention responses and at least one correct response): As shown in **Figure 3(c)**, the abstention ratio decreases over training, consistent with the intended behavior.
>
> In contrast, the binary reward leads to a collapse of abstention behavior in both groups. This demonstrates that the ternary reward scheme under GRPO adaptively generates positive or negative credits for abstention, **ultimately resulting in different learning dynamics that cannot be reproduced by simply reshaping binary rewards**.
>
>
>
> We are more than happy to incorporate additional suggestions you might have. Thanks again for your time!
>
>
>
> **References**
>
>
> [1]. Zhang et al. R-Tuning: Instructing Large Language Models to Say `I Don't Know'. NAACL 2024
>
> [2]. Yang et al. CRAG - Comprehensive RAG Benchmark. NeurIPS 2024
>
> [3]. Huang et al. ConfQA: Answer Only If You Are Confident. arXiv:2506.07309

---

> > ### Comment · Reviewer_uW9S · 2025-11-26
> >
> > Thank you for your response and for all the effort you have put into the work. Most of my concerns have now been addressed.
> >
> > Whether introducing an abstention reward counts as a true innovation may ultimately be a matter of research taste. From my personal perspective, I am not sure this alone constitutes a major novelty, although I should note that I am not an expert specifically in LLM reliability.
> >
> > Taking everything into account, I maintain my overall positive assessment of the paper. Good luck!

---

> > > ### Author Response · Authors · 2025-11-27
> > > **Thank you!**
> > >
> > > We thank the reviewer for acknowledging our efforts to address the previous concerns. We are truly grateful for the candid exchange of ideas, as well as for your thoughtful feedback and support of our work.
> > >
> > > To provide more context,  a concurrent work from OpenAI [1] also highlights that (1) hallucinations are inherently inevitable under purely accuracy-driven/binary rewarding schemes, and (2) the ability to abstain plays a critical role in mitigating hallucination, yet this has been largely neglected by most existing dominant leaderboards (see **Table 2 of [1]**). As a result, their paper argues that
> > > > *“The numerous primary evaluations must be adjusted to stop penalizing abstentions when uncertain.”*
> > >
> > > and calls for new methods that
> > > > *“can realign incentives, rewarding appropriate expressions of uncertainty rather than penalizing them”.*
> > >
> > > **In this sense, our solution TruthRL emerges as a timely and important contribution to this challenge.**
> > >
> > > We fully agree with reviewer that TruthRL is a principled method, and we’d like to note that **the value of our work lies not only in its simplicity and effectiveness, but also in the comprehensive and in-depth analyses** presented in the paper: many of them have not been explored in prior work, including systematic comparisons of learning dynamics under binary vs. ternary reward schemes, the unique dynamic credit assignment for abstention under GRPO, and studies on robustness, generalizability, and scalability. All of these efforts provide a meaningful foundation for future research.
> > >
> > > We would greatly appreciate any additional suggestions and would be glad to incorporate them into the paper. Thank you again for your time and efforts!
> > >
> > > [1] Why language models hallucinate. OpenAI. arXiv:2509.04664

---

### Meta-Review · Area_Chair_9LvM · 2026-01-09

**Summary:**

This paper proposes TruthRL, an online reinforcement learning framework that optimizes a truthfulness objective for LLMs using a ternary reward to balance correct answering, abstention, and hallucination. Reviewers agreed that the problem of hallucination and over-/under-abstention is important and timely, and that the experimental study is extensive and carefully executed across multiple benchmarks, models, and settings. However, despite these strengths, reviewers consistently raised concerns about the limited novelty of the core idea and its positioning relative to substantial prior work on abstention-aware training and reliability-oriented RL for LLMs. After considering the rebuttal, the AC recommend rejection.

**Reviewer Concerns:**

The main unresolved concern across reviewers is insufficient conceptual novelty. While the ternary reward is shown to be effective in practice and well-analyzed, similar reward structures and abstention-oriented RL or preference-based methods have been explored in prior work, and reviewers remained unconvinced that the differences articulated in the rebuttal constitute a clear methodological advance. Several reviewers also noted that the contribution is best characterized as a careful empirical and engineering study rather than a new algorithmic framework.

**Reviewer Scores:**

Reviewer scores were mixed but overall leaned to rejection, with one marginally positive review and multiple marginal or clear rejections. Importantly, no strong consensus emerged in favor of acceptance.

---

### Decision · Program_Chairs · 2026-01-26

Reject